

# Drought recorded by Ba/Ca in coastal benthic foraminifera

Inda Brinkmann[1], Christine Barras[2], Tom Jilbert[3], Tomas Naeraa[1], K. Mareike Paul[3], Magali Schweizer[2], Helena L. Filipsson[1]

[1]Department of Geology, Lund University, Sölvegatan 12, 223 62 Lund, Sweden
[2]LPG UMR CNRS 6112, University of Angers, University of Nantes, 2 bd Lavoisier 49045, Angers Cedex 01, France
[3]Aquatic Biogeochemistry Research Unit, Ecosystems and Environment Research Program, Faculty of Biological and Environmental Sciences, University of Helsinki, Viikinkaari 1, 00790 Helsinki, Finland

*Correspondence to*: I. Brinkmann (inda.brinkmann@geol.lu.se)

**Abstract.** Increasing occurrences of extreme weather events, such as the 2018 drought over northern Europe, are a concerning issue under global climate change. High resolution archives of natural hydroclimate proxies, such as rapidly accumulating sediments containing biogenic carbonates, offer the potential to investigate the frequency and mechanisms of such events in the past. Droughts alter the barium (Ba) concentration of near-continent seawater through the reduction in Ba input from terrestrial runoff, which in turn may be recorded as changes in the chemical composition (Ba/Ca) of foraminiferal calcium carbonates accumulating in sediments. However, so far the use Ba/Ca as a discharge indicator has been restricted to planktonic foraminifera, despite the high relative abundance of benthic species in coastal, shallow-water sites. Moreover, benthic foraminiferal Ba/Ca has mainly been used in open ocean records as a proxy for paleo-productivity. Here we report on a new geochemical dataset measured from living (CTG-labelled) benthic foraminiferal species to investigate the capability of benthic Ba/Ca to record changes in river runoff over a gradient of contrasting hydroclimatic conditions. Individual foraminifera (*Bulimina marginata*, *Nonionellina labradorica*) were analyzed by laser-ablation ICP-MS over a seasonal and spatial gradient within Gullmar Fjord, Swedish west coast during 2018–2019. The results are compared to an extensive meteorological and hydrological data set, as well as sediment and pore-water geochemistry. Benthic foraminiferal Ba/Ca correlates significantly to riverine runoff, however, the signals contain both spatial trends with distance to Ba-source, and species-specific influences such as micro-habitat preferences. We deduce that shallow-infaunal foraminifera are especially suitable as proxy for terrestrial Ba input and discuss the potential influence of water-column and pore-water Ba cycling. While distance to Ba-source, water depth, pore-water geochemistry, and species-specific effects need to be considered in interpreting the data, our results



demonstrate confidence in the use of Ba/Ca of benthic foraminifera from near-continent records as proxy for past riverine discharge and to identify periods of drought.

## 1 Introduction

Extreme weather events, such as drought and heatwaves, have a devastating impact globally (e.g., Sternberg, 2011). Droughts occur through an interplay of below-average precipitation, high evapotranspiration, and hydrological preconditions, such as soil moisture and water storage capacity (e.g., Tallaksen and Van Lanen, 2004). Alone in the 21st century, Europe was affected by a series of exceptionally severe drought events (e.g., Bastos et al., 2020a). With anthropogenic global warming accelerating the internal processes driving hydroclimate variability, droughts are expected to keep increasing in frequency and severeness

in the future (e.g., Cook et al., 2020). Thus, it is imperative to document droughts pre-dating anthropogenically-forced climate change to define a baseline of natural hydroclimatic variability.

Hydrological drought, defined as deficiency in water supply including below-normal river discharge, is one of the main comprehensive indicators for accumulated hydrological responses to a reduction in precipitation on a wide spatial and temporal scale (e.g., Tallaksen and Van Lanen, 2004). A range of natural archives for historical droughts exist in the terrestrial

environment, including dendrochronological, ice core- and speleothem-based proxy records (e.g., Steiger et al., 2018). In aquatic systems, sediments offer the potential to record hydrological conditions on land through changing fluxes of materials across the land-water interface. One potential sediment proxy capable of quantifying hydrological drought is the barium to calcium ratio (Ba/Ca) in biogenic carbonates formed in the coastal marine environment. Barium in seawater is dominantly sourced from fluvial input (e.g., Wolgemuth and Broecker, 1970), and proportionally incorporated in biogenic calcium

carbonates as a substitute for the $Ca^{2+}$ ion (e.g., de Nooijer et al., 2017). Hence, changes in fluvial Ba inputs may be expected to influence carbonate Ba/Ca in the near-shore environment.

While Ba/Ca is applied for paleoclimatic reconstructions, the focus so far has been on corals in tropical environments (e.g., Saha et al., 2018) and planktonic foraminifera (e.g., Bahr et al., 2013; Mojtahid et al., 2019). Moreover, Ba/Ca of benthic foraminifera is dominantly interpreted as paleo-productivity proxy (e.g., Mojtahid et al., 2019; Ní Fhlaithearta et al., 2010),

due to the nutrient-like cycling of Ba in seawater (e.g., Dehairs et al., 1980; Paytan and Griffith, 2007). Benthic foraminifera

are ubiquitous in coastal sediments and concurrently coastal sediment records are important paleoclimate archives due to their high temporal resolution and transitional position between the terrestrial and marine realm (review by Howe et al., 2010). While rare applications of benthic foraminiferal Ba/Ca as runoff indicator exist (Groeneveld et al., 2018; Ni et al., 2020), studies focusing on living benthic foraminifera are distinctly lacking. In-situ, core-top approaches with living foraminifera are,

however, essential to calibrate a proxy and create a reliable baseline of empirical observations by correcting for species-specific biases and seasonal-scale mechanisms influencing Ba/Ca in a coastal environment.

The drought and heatwave of 2018 that affected Northern and Central Europe, including Scandinavia, was exceptional in its severeness (e.g., Peters et al., 2020), and was followed by a warm and wet year in 2019 (Swedish Meteorological and Hydrological Institute, SMHI). This contrast in precipitation between 2018 and 2019 presents a unique opportunity to study

the response of benthic foraminiferal trace-elemental concentrations to changes in riverine discharge. Here we test the hypothesis of benthic foraminiferal Ba/Ca as a tracer for riverine discharge, assess the potential of identifying drought events in sediment foraminiferal records, and evaluate the broader implications for the interpretation of benthic Ba/Ca in paleo-studies in coastal environments.

**2 Site description: Gullmar Fjord, Swedish west coast**

Gullmar Fjord (GF; ca. 28 km by 1–2 km; Fig. 1) has a maximum water depth of 120 m, located centrally at Alsbäck, and a sill depth of 43 m towards the Skagerrak. The sediments in the fjord's deep basin are comprised of silty clays with a low sand content and an organic carbon content of approximately 3 % in surface sediments (e.g., Gustafsson and Nordberg, 2001). Phytoplankton blooms occur typically biannually in spring and autumn (dominated by diatoms and dinoflagellates, respectively; e.g., Gustafsson and Nordberg, 2001).

The river Örekilsälven is the main contributor of freshwater to the fjord (annual average discharge of 24 m³/s vs. 0.6–1.0 m³/s inflow from catchment north and south of GF) and represents > 80 % of the fjord's catchment area (1338 km² out of ~ 1600 km²; Fig. 1). The annual and seasonal discharge is proportional to the respective hinterland precipitation. Örekilsälven's inflow contributes to the fjord's brackish surface water layer (~ 0.5 m; Fig. 1).

Further, the fjord's hydrography is governed by the water exchange with Skagerrak (North Sea) and Kattegat (Baltic) (e.g.,

Arneborg, 2004). While water masses exchange more readily above sill-level with Kattegat-waters, the deep basin (> 50 m)

flushes approximately annually, typically between January and April, and the ventilation events are driven by wind direction
and the winter NAO pattern (e.g., Filipsson and Nordberg, 2004). Except for periods of water renewals, the fjord's waters are
stratified in layers of distinct salinity and renewal times (e.g., Arneborg, 2004; compare Fig. 1).

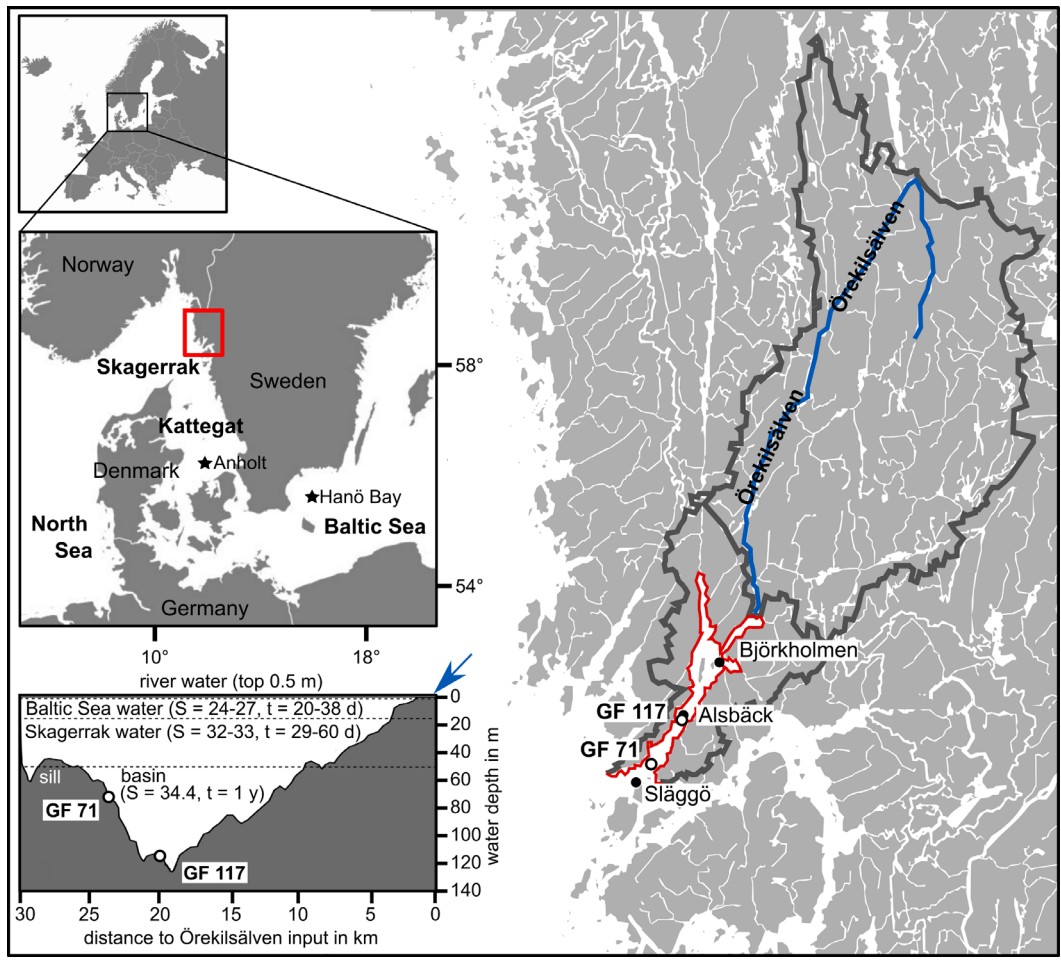

**Figure 1: Location of the study area within Europe and on the west coast of Sweden (marked by red rectangle) giving an overview
of Gullmar Fjord and Örekilsälven, including cumulative catchment area (outlined in dark grey). The study sites GF 71 and GF
117, and environmental monitoring sites Alsbäck, Björkholmen, and Släggö are indicated. A transect of Gullmar Fjord with water
masses of different sources is given (S = salinity, t = typical residence time in days (d) or years (y); after Arneborg, 2004), with
indication of Örekilsälven input (blue arrow). Maps modified after SMHI and Groeneveld et al. (2018). Color online.**

## 3 Material and methods

### 3.1 Sampling

Our study is based on three seasonal sampling campaigns (September 2018, February and June 2019) in Gullmar Fjord, Swedish west coast (Table 1). During each campaign two sites were sampled, GF 71 (water depth ~ 70 mbsl) and GF 117 (~ 116 mbsl). A CTD with $O_2$-probe and Niskin water-collection system was deployed to record temperature, dissolved oxygen concentration ($[O_2]$), and salinity (reported using the practical salinity scale) throughout the water column and to collect bottom-water. Bottom-water was sampled in bottles of 25–40 mL and refrigerated until further analyses. Sediment cores were collected with a GEMAX® twin-barrel corer (modified Gemini corer, 9 cm diameter, from Oy Kart AB, Finland) in two casts. For foraminiferal analyses, the top one cm of two duplicate cores from the first casts were sliced in 0.5 cm sections (ca. 32 ml sediment; 0–0.5 cm, 0.5–1.0 cm). Each sample was placed into an HDPE (High Density Polyethylene) bottle with roughly equal amounts of ambient bottom water and 36 µl of CellTracker™ Green (CTG) CMFDA dye (5-chloromethylfluorescein diacetate; with dimethyl sulfoxide (DMSO); both ThermoFisher Scientific) to reach a final CTG concentration of 1 µM. The samples were incubated at 6 ºC (in-situ bottom-water temperature 6.5–7.1 ºC) for 10 hours for living cells to take up the CTG epifluorescent tracer for later observation (e.g., Bernhard et al., 2006), and thereafter preserved in 40 ml of ethanol.

At site GF 117, one set of duplicate cores was sampled for bottom-, pore-water, and solid-phase geochemical analyses. During each campaign, the first core was used for bottom and pore-water sampling using Rhizons™ at 2 cm vertical resolution. The samples were collected into 10 mL polyethylene syringes through pre-drilled holes (diameter 4 mm; e.g., Jokinen et al., 2020) immediately after core retrieval. Within a few hours after retrieval, all water samples were transferred into 15 mL polypropylene centrifuge tubes and acidified to 1M $HNO_3$ for elemental analyses.

The second core from each pair was sampled for solid-phase parameters. In September 2018, the second core was immediately sampled on deck after core retrieval. In February and June 2019, the second core was sampled a few hours after core retrieval inside a cool room. Each core was sliced at 1 cm intervals (0–10 cm) and 2 cm intervals (10–30 cm). Each sediment slice was transferred into plastic bags, which were carefully sealed under water, and transferred into gas tight glass jars. The jars were flushed with $N_2$ and stored dark at 4°C prior to further processing to avoid oxidation of the samples. In preparation for solid-phases analyses, each wet sediment slice was subsampled inside a $N_2$-flushed glove bag and frozen for 24h at -20°C.





110 Subsequently, each sample was freeze-dried for 48h, homogenized, and weighed in between each step to determine the water
[g] and salt contents [g], and porosity [cm$^3$ cm$^{-3}$] using the bottom water salinity and the assumed solid-phase density of 2.65
g cm$^{-3}$ (Burdige, 2006). The water and salt contents were used to determine the salt-free weight of the dry sediment, which is
needed to correct the solid-phase elemental concentrations for salt-dilution.

**Table 1: Sampling sites and number of analyzed specimens per species and sediment interval for each site and sampling occasion (\***
115 **marks samples compiled from two duplicate cores), before and after quality control of measurements. Median Ba/Ca with median**
**average deviation (MAD) of n-chambers in µmol/mol per species, site, and season (number of data points, i.e., analyzed specimens,**
**indicated in brackets) for top 1 cm sediment.**

| Site (Water depth) Position | Species | Sampling date | Number of analyzed specimens | | After quality control (n-chamber spots) | | Median Ba/Ca with MAD (µmol/mol) |
|---|---|---|---|---|---|---|---|
| | | | **0–0.5 cm** | **0.5–1.0 cm** | **0–0.5 cm** | **0.5–1.0 cm** | **0–1 cm** |
| **GF 117 inner fjord** (115–117 m) 58º19.695'N, 11º33.147'E | *N. labradorica* | 17 Sept 2018 | 9* | 11 | 8 | 9 | 2.77 ± 1.13 (17) |
| | | 24 Feb 2019 | 10 | 8 | 8 | 7 | 3.43 ± 1.53 (15) |
| | | 11 June 2019 | 7 | 9 | 7 | 6 | 2.89 ± 0.71 (13) |
| | *B. marginata* | 17 Sept 2018 | 17* | 10 | 15 | 9 | 1.82 ± 1.04 (24) |
| | | 24 Feb 2019 | 9 | 8 | 4 | 5 | 6.84 ± 1.98 (9) |
| | | 11 June 2019 | 11 | 10 | 10 | 10 | 6.63 ± 2.25 (20) |
| **GF 71 outer fjord** (69–71 m) 58º17.116'N, 11º30.546'E | *N. labradorica* | 17 Sept 2018 | 10 | 10 | 9 | 9 | 1.97 ± 0.26 (18) |
| | | 24 Feb 2019 | 10 | 10 | 8 | 10 | 4.33 ± 1.07 (18) |
| | | 11 June 2019 | 11 | 11 | 10 | 11 | 4.10 ± 0.72 (21) |
| | *B. marginata* | 17 Sept 2018 | 13 | 10 | 4 | 8 | 0.55 ± 0.13 (12) |
| | | 24 Feb 2019 | 11 | 11 | 8 | 7 | 2.82 ± 1.09 (15) |
| | | 11 June 2019 | 10 | 11 | 8 | 9 | 2.19 ± 1.15 (17) |

## 3.2 Geochemical analyses of benthic foraminifera

Living (CTG-labelled) specimens of *Bulimina marginata* d'Orbigny, 1826 and *Nonionellina labradorica* (Dawson, 1960) were selected for laser-ablation (LA) ICP-MS trace element analysis, according to their availability in each sample (i.e., abundance). To extract specimens the samples were washed over a set of sieves (63, 100 μm) and the > 100 μm fraction was wet-picked under epifluorescent light (Nikon SMZ 1500 with Nikon Intensilight C-HGFI). When a single core produced an insufficient number of specimens for analyses, specimens from the duplicate cores were also analyzed. An average of 10 living specimens – identified by their CTG-label – per sediment interval and site were selected for geochemical analyses (Table 1). The tests were treated with NaOCl (5%) for 2–5 hours to remove organic materials incl. cytoplasm (e.g., Mashiotta et al., 1999), following a triple rinse with Milli-Q (Milli-Q Integral, EMD Millipore Corporation, Billerica, MA, USA) water and dried at room-temperature.

Laser Ablation ICP-MS analyses were carried out at the Dept. of Geology, Lund University, Sweden, using a Bruker Aurora Elite (quadrupole) ICP-MS and a 193 nm Cetac Analyte G2 excimer laser installed with a two volume HelEx2 sample cell. Helium was used as carrier gas (approximately 0.8 l/min) in combination with a down-stream addition of Argon (approximately 0.95 l/min) after the sample chamber. The counts of the isotope masses $^{138}$Ba, as well as $^{27}$Al, $^{43}$Ca, $^{55}$Mn and $^{66}$Zn were acquired at 5 Hz (i.e., laser shots per second). To obtain a stable isotope signal a "squid" was inserted downstream of the sample chamber. The scan time of the ICP-MS was synchronized with the laser repetition rate to minimize artificial fluctuations of the isotope signals. Spot sizes were adapted to chamber sizes of the investigated species (75x75 μm, 55x75 μm, 50x70 μm), but kept constant within individual sessions and applied to both samples and standard material. Previous studies indicate that using variable spot sizes does not influence elemental fractionation during laser-ablation measurements on foraminiferal tests (e.g., Eggins et al., 1998). The samples were analyzed at an energy density of 1 J/cm$^2$ to avoid breaking of foraminiferal tests and obtain a longer ablation signal. The conditions were mimicked for the secondary carbonate standards. Glass standards, U.S. National Institute of Standards and Technology SRM NIST610 and 612 (see below), were analyzed at 3 J/cm$^2$ for optimized ablation efficiency. The LA-ICP-MS was operated in manual mode for sample analyses. Elemental baseline levels before each spot analysis were measured for at least 30 sec.

NIST610 was used for instrument tuning to ascertain high and stable signal counts on the studied elements, low oxide
production (< 0.5 % monitoring $^{238}U/^{238}U^{16}O$ and $^{232}Th/^{232}Th^{16}O$), and low elemental fractionation ($^{232}Th/^{238}U$ ratios close to
1). NIST610 was used as external calibration material with GeoReM (Geological and Environmental Reference Materials;
Jochum et al., 2005) composition values (via http://georem.mpch-mainz.gwdg.de), giving an average internal relative precision
for Ba of 4.4 % (based on the relative standard deviation of raw data (cps) per analysis session; Table S1), and to correct for
instrumental drift. Primary standard NIST 610 and secondary standards, NIST612, MACS-3 (MicroAnalytical Carbonate
Standard, United States Geological Survey, 2012; Jochum et al., 2012), JCt-1 and JCp-1 (AIST Japan calcium carbonate
pellets) were run at the beginning and end of each session and the data used to monitor the quality of the analyses (Table S1).
A selection of these – NIST 610, MACS-3, JCp-1 – were additionally run after each 10–15 sample spots.

For each specimen, the final (n), penultimate (n-1), and antepenultimate (n-2) chambers (as far as possible), were targeted and
ablated from the outside to inside of the wall. For *B. marginata* also the signal of the initial chambers was measured with a
laser-spot covering proloculus and following chambers and ablating through these chambers (referred to as 'p'). We consider
this to represent a bulk signal of an extended growth period, including the primary signal of the juvenile chambers as well
subsequently added layers of calcite with each new chamber addition (e.g., Erez, 2003).

### 3.3 Benthic foraminiferal trace-element data processing

Raw counts were converted to elemental concentrations (ppm) with the software package Igor Pro 6.37 with Iolite v3.5 (Paton
et al., 2011). The element ratios were calculated based on the average concentration measured during the ablation period of
the primary test calcium carbonate. This signal interval was determined through manual examination of each trace element
profile. Constant counts of $^{43}Ca$ were used to find integration intervals of calcite ablation. Intervals of enriched signals of $^{27}Al$,
$^{55}Mn$ and/or $^{66}Zn$, indicative of secondary surface coatings and/or contaminations, were excluded. Such were typically
measured at the beginning and/or end of each profile (corresponding to the outer and inner test wall surfaces, respectively),
with relatively constant raw elemental counts intermediate (see also e.g., Ni et al., 2020). Trace elemental ratios (TE/Ca) were
calculated with the assumption of the TEs being bound as carbonate (assuming 40% wt Ca in the $CaCO_3$ matrix) and
normalized to calcium.


In total, 486 ablation profiles (> 15 data points of the ICP-MS per signal) were performed on chambers (n, n-1, n-2 and p+) of *B. marginata* and 346 on chambers (n, n-1, n-2) of *N. labradorica* specimens and evaluated according to the following criteria

(i.e. quality control): (1) [Ba] higher than Limit of Quantification (LOQ = baseline signal + $10 \times \sigma$; $\sigma$ = standard deviation of baseline signal) (9 signals), (2) exclusion of ablation signals with potential clay contamination (i.e., Al/Ca > 0.4 mmol/mol; Wit et al., 2010) (30 signals), (3) application of a 95% confidence limit on Ba/Ca, for each species, site and season, to remove outliers (i.e., > 2 * standard deviation) (39 signals). In total, 78 of 832 (9.4 %) ablation profiles were rejected from further analyses.


From the benthic foraminiferal Ba/Ca values the median and MAD was calculated per species, season, and site, to limit the influence of extreme outliers. As no systematic differences between sediment intervals were observed, Ba/Ca data were pooled (Welch ANOVA, Table S2). If not indicated otherwise, only data on n-chambers were considered, assuming these were precipitated in close temporal proximity to the time of sampling.

**3.4 Pore-water and sediment trace-element analyses**

The acidified bottom-, and pore-water samples were analyzed for elemental concentrations of Ba by Inductively Coupled Plasma-Mass Spectrometry (ICP-MS; Thermo Scientific XSeries 2; Utrecht University, Department of Earth Sciences) and of manganese (Mn), iron (Fe), and sulfur (S) by Inductively Coupled Plasma-Optical Emission Spectrometry (ICP-OES; Thermo Scientific iCAP 6000; Helsinki University, Ecosystems and Environment Research Program). For the determination of total

solid-phase Ba, Mn, and Fe concentrations, an aliquot of 100-125 mg of freeze-dried and powdered sediment was digested in 2.5 mL solution of 3:2 $HClO_4$ (70%) and $HNO_3$ (65%) and HF (40%). The Teflon vessels were left with closed lids overnight under a fume hood with water trap on a hotplate at 90 °C. After 12h, the lids were removed, and the extracts were heated to 140 °C to evaporate the acids. The evaporate was re-dissolved in 25 mL HNO3 (4.5%) and left overnight on a hotplate at 90 °C. The next day, the Teflon vessels were weighed again to determine the dilution of the final solutions. Subsequently, the

sample residues were analyzed by ICP-MS (Ba) at Utrecht University and by ICP-OES (Mn, Fe) at the University of Helsinki. Based on sediment sample duplicates the average analytical uncertainties were 6 ppm (Ba), 3.5 ppm (Mn), and 18.6 ppm (Fe).

## 3.5 Meteorological and hydrological data

All meteorological, hydrographical, and hydrological data of the Swedish west coast, the sill fjord, and its catchment area are publicly available from SMHI (http://www.smhi.se). We used the SHARK hydrographic database (https://sharkweb.smhi.se)

to obtain monthly data on salinity, $[O_2]$, and/or chlorophyll-a across 2018 and 2019 from three fjord stations Släggö (58º25.98'N, 11º43.57'E, 70 mbsl), Alsbäck (58º19.40'N, 11º32.80'E, 120 mbsl) and Björkholmen (58º23.26'N, 11º37.60'E, 50 mbsl) (Fig. 1), of which Alsbäck is closely located to GF 117. Daily runoff and sediment load data were collected from the hydrological database 'vattenweb' (http://vattenweb.smhi.se/, last access: February 2021), partly based on the S-HYPE (Hydrological Predictions for the Environment) model maintained by SMHI. For both hydrological and meteorological data,

the averages were calculated for 30-, 60-, 90 and 120-day periods preceding the sampling occasions directly or by one month, respectively (i.e., four periods ending on and four period ending one month prior to the sampling dates).

## 3.6 Statistical analyses

All statistical analyses were performed with the software package PAST 4.05 (Hammer et al., 2001), considering a p-value of < 0.05 as significant. Since parts of the data were non-normally distributed (Shapiro-Wilk's test) and heteroscedastic (Bartlett's

test), we performed ANOVA Welch or Kruskal Wallis tests associated with Mann Whitney posthoc tests to compare Ba/Ca between seasons, sediment intervals, species, and chambers (non-parametric tests). To determine strength and direction of monotonic relationships between categorial and continuous variables (e.g., chamber number and Ba/Ca) Spearman rank correlations were tested. We used Pearson's r to determine the relationship between two continuous variables (e.g., Ba/Ca and river discharge; dissolved TE concentrations), using normalized data if testing data across several sites and species (0–1

scaling).

## 4 Results and discussion

### 4.1 Contrasting hydroclimatic conditions in 2018/2019 and Ba/Ca shift in benthic foraminifera

In the year 2018 north-western Europe experienced severe heat and drought, ascribed to an interplaying Rossby Wave-7 circulation and positive North Atlantic Oscillation index (NAO) across the Northern Hemisphere (e.g., Kornhuber et al., 2019).





The occurrence of this large-scale synoptic weather pattern has increased within the last three decades, being responsible also

for the 2003, 2006, 2012, and 2015 drought events, in response to anthropogenic global warming (e.g., Kornhuber et al., 2019).

In Sweden, the drought was expressed as above-normal mean air temperatures and below-normal precipitation, and distinctly

imprinted in the seasonal runoff conditions across the country (Fig. 2a; SMHI).

While the hydrological cycle naturally experiences seasonal variability – high runoff in winter, spring, and autumn (150 mm

each), low in summer (50 mm) – ,in 2018 already the spring was exceptionally dry. The very dry spring conditions likely

contributed to amplifying the following summer drought (Bastos et al., 2020b), with summer and autumn precipitation being

50–75% of the 1961–1990 average (southwestern Sweden).

On the contrary, the year 2019 registered above-average annual precipitation (120–130 % in southwestern Sweden), although

strongly varying monthly (e.g., record-low in April, highest precipitation in May), and was warmer-than-average both in

Sweden and globally (Dunn et al., 2020).

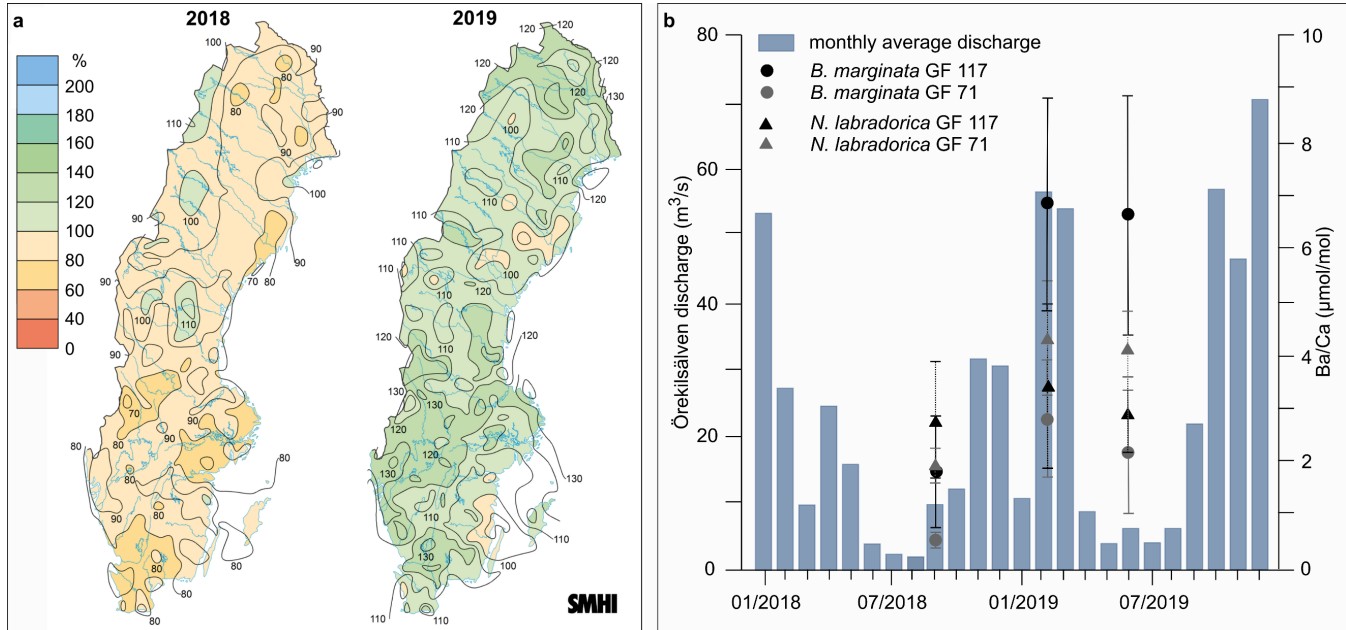

**Figure 2: (a) Maps of Sweden showing annual average runoff in 2018 and 2019 compared to reference period 1961–1990 (SMHI); (b) Average monthly river discharge of Örekilsälven across 2018 and 2019 with median Ba/Ca with MAD per investigated species (n-chambers) and site indicated. Color online.**






In the study area the contrasting hydroclimate conditions of 2018 and 2019 expressed in annual average discharges of Örekilsälven to GF of 18.5 and 29.0 m³/s, respectively (Fig. 2). Coinciding, the benthic foraminiferal Ba/Ca shifted significantly in most samples, corresponding to an increase of up to 5-times (Table 1; Fig. 2b, 3, S1).

Correlations of foraminiferal Ba/Ca with the shift in hydroclimatic conditions of the consecutive years imply Ba/Ca

representing the discharge 60–90 days prior to sampling (r = 0.36–0.40), or discharge conditions preceding the Ba/Ca record by one month (r = 0.46–0.48; testing Ba/Ca across species and sites; Table S3). The lagging aims to account for the uncertainties in the exact timing of foraminiferal growth, as well as the potential lag time in the transport of riverine Ba to coastal sediments. The lag is probably most applicable to the foraminiferal samples collected in 02/2019, assuming the majority of foraminiferal calcite was precipitated in connection to the preceding autumn phytoplankton bloom (e.g., Gustafsson and

Nordberg, 2001; Fig. S2). Temporal differences in reproduction and growth of the species could explain the differing strength of correlation to the observed hydroclimatic conditions, as well as species-specific intrinsic factors that can influence benthic foraminiferal Ba/Ca (further discussed in Sect. 4.3).

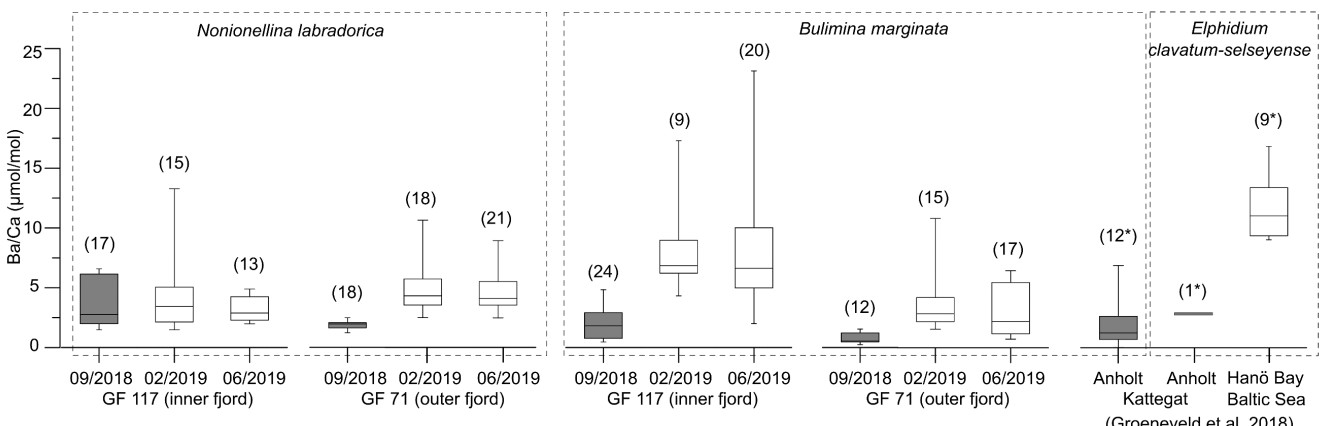

**Figure 3: Box- and whisker plots of benthic foraminiferal Ba/Ca from Gullmar Fjord (this study), Anholt (Kattegat) and Hanö Bay**

**(Baltic Sea) (Groeneveld et al., 2018). Grey boxes represent samples from conditions without direct influence of terrestrial Ba addition, white boxes terrestrial-influenced samples. Number of analyzed specimens (for LA-ICP-MS analyses) or bulk samples (of > 8 specimens each, ICP-OES analyses, marked by *) given in brackets. Note Ba/Ca being derived from different species (marked by dashed rectangle).**


The range of Ba/Ca across 2018/19 varies between the inner-fjord and the outer-fjord site in all species (1–4-times vs. 2–5 times, respectively), with generally lower Ba/Ca in the outer-fjord site during 2018 (Fig. 3). We hypothesize that the outer-fjord site experiences larger differences between periods of contrasting influence of Ba-rich river- versus low-Ba seawater of Kattegat and Skagerrak. This is plausible as the influence of Ba addition by Örekilsälven discharge diminishes, and the frequency of water exchange events increases, from the inner to outer fjord (Fig. S3). The fjords topography could be an

additional factor, with the deep basin acting as a Ba sink. While the effects are minor compared to those of prolonged drought periods, these spatial factors should be of consideration when inferring hydroclimatic gradients from foraminiferal Ba/Ca.

The Ba/Ca signals of 2018 compared well to values expected for environments with negligible terrestrial influence (e.g., Anholt site in Kattegat see Fig. 1, compilation of data from this study and Groeneveld et al., 2018 in Fig. 3), thus are anomalously low considering the close spatial proximity to a river mouth (< 25 km; Fig. 1) and seasonally restricted water exchanges with the

open sea (Fig. S3). On the other hand, Ba/Ca of the 2019 fjord samples range distinctly higher, consistent with the expectation for coastal environments such as the Baltic Sea (Fig. 1 for location; Fig. 3).

We deduce that the drought prevailing in spring and summer 2018 caused a distinct lack of riverine input, interrupting terrestrial Ba addition to the fjord. The results imply that benthic foraminiferal Ba/Ca in the coastal environment responds on short timescales to the prevailing hydrological conditions on the adjacent continent.

**4.2 Sediment evidence for Ba cycling in Gullmar Fjord**

Barium is incorporated into foraminiferal tests proportionally to ambient concentrations (e.g., de Nooijer et al., 2017). Hence, contrasting $Ba^{2+}$ availability in the foraminifera's microenvironment is a prerequisite for the observed benthic Ba/Ca signals. For planktonic foraminifera it has been shown that Ba incorporation is temperature- and salinity-independent (e.g., Hönisch et al., 2011). As both parameters were stable in the fjord (T = 6.7 ± 0.2 ºC, S = 34.4 ± 0.1), an influence on benthic Ba/Ca can be

excluded. Bottom-waters stayed oxygenated, albeit over a large gradient (ambient $[O_2]$ during sampling = 70–217 µmol/l; Fig. S3), and $[O_2]$ shows no correlation to benthic Ba/Ca. This implies that $Ba^{2+}$ concentrations close to the sediment-water interface must have changed between the sampling moments.

The expected difference in $Ba^{2+}$ availability between 2018 and 2019, however, is not clearly supported by the Ba sediment profiles obtained during sampling (Fig. 4; based on GF 117). Sedimentary Ba contents range from 3.11–3.53 µmol/g, indicating





a strong enrichment relative to sediments of the Skagerrak (ca. 0.41 µmol/g particulate Ba; Lepland et al., 2000), confirming

a near-shore accumulation due to terrigenous inputs. However, no significant changes were recorded in the surface sediment

Ba concentration. Meanwhile, pore-water Ba profiles indicate diagenetic remobilization of Ba at depth in the sediment column.

Values in excess of 1 µmol/l $Ba^{2+}$ at 20 cm depth – distinctly higher than marine water column values (Kattegat [Ba] = 0.02–

0.06 µmol/l, Groeneveld et al., 2018; global open ocean surface [Ba] = 0.04–0.05 µmol/l, Hsieh and Henderson, 2017) –

support the concept of Ba release into pore-waters fueling potential uptake into benthic foraminifera. However, the pore-water

profiles also do not show significant changes between sampling years and most of the enrichment of $Ba^{2+}$ in porewaters is

observed deeper in the sediment column.

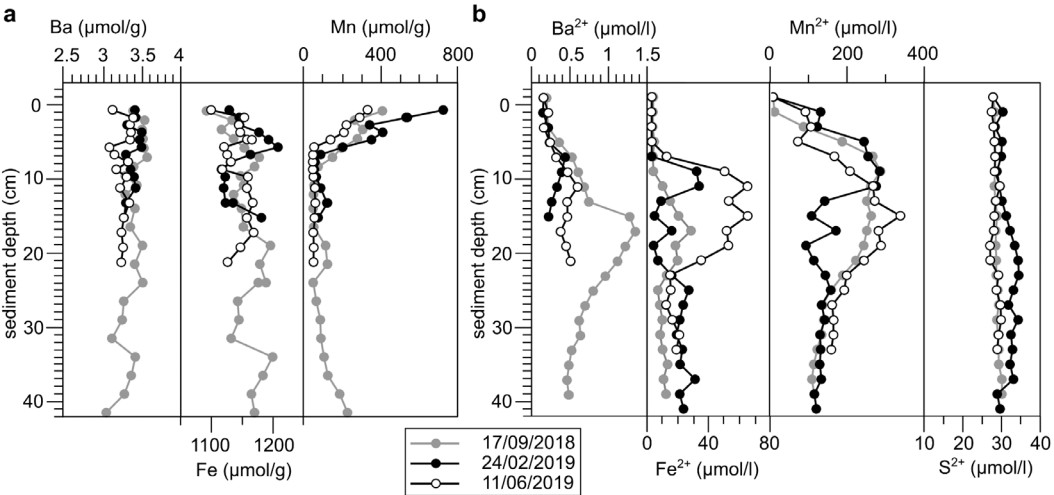

**Figure 4: (a) Solid-phase and (b) dissolved trace element profiles of site GF 117 at each sampling occasion.**


To explain the observed changes in foraminiferal Ba/Ca between 2018 and 2019 in the absence of whole-core sediment or

pore-water geochemical evidence, we infer that rapid sub-annual processes at the sediment-water interface must be responsible.

With sedimentation rates at the study sites being 0.7–0.9 cm/year (Filipsson and Nordberg, 2004), only millimeters of new

sediment material are expected to accumulate between the sampling occasions. Sub-annual changes in reactive Ba supply

within this material, however, may dictate pore-water Ba release close to the sediment-water interface and uptake into benthic

foraminifera. Such changes are unlikely to be captured in the pore-water profiles obtained at 2 cm resolution on the three

sampling occasions. We infer therefore that during wet years such as 2019, reactive Ba is transported into the fjord in short-

timescale pulses directed by continental rainfall and riverine discharge events, resulting in transient high fluxes of reactive Ba, which can be recorded by benthic foraminiferal Ba/Ca. Such temporally restricted fluxes may also explain the high relative

standard deviation of Ba/Ca (in n chambers of *B. marginata* = 46–71 %; *N. labradorica* = 17–69 %). While chamber precipitation in foraminifera is generally completed within a few hours (e.g., de Nooijer et al., 2009), population signals represent cumulative seasonal trends in terrestrial Ba input and benthic availability. In the following we explore potential sources of reactive Ba.

### 4.2.1 Potential sources of short-timescale changes in Ba/Ca: biogenic barite cycling

In marine environments Ba can be associated with various particulate phases, including detrital or 'biogenic' barite ($BaSO_4$), and adsorbed to organic matter, carbonates, or metal oxides (e.g., Coffey et al., 1997; Dehairs et al., 1980; McManus et al., 1998). Of these, biogenic barite is the most commonly studied and therefore we must consider the potential of biogenic barite cycling to influence the observed signals in foraminiferal Ba/Ca.

In open-ocean settings primary productivity acts as the main facilitator of downward Ba transport: dissolved Ba is adsorbed or

bound to primary producers and forms biogenic barite upon organic matter degradation in the water column (e.g., Dehairs et al., 1980; Martinez-Ruiz et al., 2019). Thereby, a Ba-productivity relationship develops that is frequently used to infer paleo-productivity changes from benthic foraminiferal Ba/Ca (e.g., Mojtahid et al., 2019; Ní Fhlaithearta et al., 2010). For example, shifts from low to high paleo-productivity periods reflect in benthic Ba/Ca increases by 2–3-times in the Mediterranean (pre- and during sapropel S1; Mojtahid et al., 2019; Ní Fhlaithearta et al., 2010), and <1.5 in the Bahama Banks and Caribbean Sea

(Last Glacial Maximum to deglaciation; Hall and Chan, 2004). However, this relationship is a function of water depth due to barite crystal growth- and settling rates and cannot be reliably extended to shallow-water deposits as studied here (< 1000 m water depth; e.g., Von Breymann et al., 1992). Still, episodic removal of $Ba^{2+}$ by barite formation in association with phytoplankton blooms is known from estuaries, although typically showing rapid Ba regeneration with riverine discharge surges (e.g., Stecher and Kogut, 1999).

A theoretical mechanism for short-timescale release of Ba from biogenic barite in sediments is variable porewater sulfate undersaturation (e.g., McManus et al., 1998). However, no evidence for this mechanism is observed in the fjord. Pore-water total S concentrations, primarily reflecting sulfate, are stable throughout the sediment column (Fig. 4b). Organo-clastic sulfate

EGU Open Access

reduction (e.g., McManus et al., 1998) is primarily activated once more energetically beneficial electron acceptors are consumed (oxygen, nitrogen, manganese (Mn), iron (Fe); e.g., Froelich et al., 1979). In the sediments of GF the anaerobic

reduction of Mn and Fe (oxyhydr)oxides (hereafter referred to as Mn or Fe oxides) dominates, as evidenced by pore-water Mn and Fe enrichments (down to ca. 12.5–25 cm depth; Fig. 4b). The fjord's high sedimentary Mn content makes sulfate reduction and diagenetic $Ba^{2+}$ release within the habitat of benthic foraminifera generally unlikely (e.g., Vandieken et al., 2012). Hence, both biogenic and detrital barite can be considered as unreactive in this setting.

### 4.2.2 Ba and metal oxide cycling

In transition zones between fluvial and marine environments, such as GF, estuarine mixing behavior modulates Ba addition from the continents through the desorption of $Ba^{2+}$ from suspended matter of fluvial origin (terrestrial aluminosilicates and other detrital material) by cation exchange at low to medium salinities (e.g., Coffey et al., 1997; Hanor and Chan, 1977). However, Ba may then become associated with Fe and Mn oxides (e.g., Coffey et al., 1997), due to the oxides' affinity for Ba and its chemical analogues (e.g., Balistrieri and Murray, 1986). This association potentially allows Ba to be sedimented in the

near-shore environment, providing a mechanism for short-term changes in Ba supply to the seafloor.

An overall coupling of Ba to Mn (and Fe) cycling in the GF sediments is supported by the pore-water profiles showing coinciding pore-water maxima of these elements (Fig. 4) and positive inter-element correlations (Fig. S4). We propose that short-timescale pulses of Fe and Mn oxide sedimentation in association with discharge events also lead to rapid release of Ba into solution at the seafloor as oxides are utilized in the remineralization of fresh organic matter. Based on experimental

estimates of Fe/Mn oxide particle settling velocity in the Baltic oxides shuttle through the water-column in the order of weeks, considering the water depths of the study sites (Glockzin et al., 2014). This constitutes a transport mechanism with a short temporal delay between surface Ba addition and foraminiferal incorporation at depth that is in line with the correlation of Ba/Ca to riverine fluxes strengthening with increasing time between sampling and river discharge-period (see Sect. 4.1; Table S3).

The influence of $Ba^{2+}$ liberated from recently accumulated Fe and Mn oxides appears to decrease below the water-sediment interface, where $Ba^{2+}$ availability is dictated more by processes deeper in the sediment column. Barium is remobilized at depth in the sediment column during oxide reduction, as indicated by the upward diffusion of $Ba^{2+}$ from below ca. 10 cm depth (Fig.

4b). This liberation of $Ba^{2+}$ from Mn oxides appears to be constant, as well as negligible in magnitude compared to the terrigenous sedimentary Ba background signal, showing the dominance of non-reactive Ba in the system (Fig. 4a). Therefore,

shallow-infaunal species such as *B. marginata* are expected to be exposed to greater variability in $Ba^{2+}$ availability than deep infaunal species.

### 4.3 Proxy potential of *N. labradorica* and *B. marginata*

Ba/Ca of *N. labradorica* and *B. marginata* are significantly different for most seasons at each site (Table S4; Welch ANOVA). Partly, this may be an effect of species-specific differences in trace element incorporation, which has been documented for

several taxa of foraminifera including *Bulimina* and *Nonionellina* (e.g., Barras et al., 2018; de Nooijer et al., 2017; Koho et al., 2017).

While *B. marginata*'s Ba/Ca reliably records fluctuations in terrestrial-derived Ba input, *N. labradorica* shows seasonal shifts in Ba/Ca only in the outer-fjord site. Against expectation, Ba/Ca of the inner-fjord site does not deviate substantially from values recorded during the drought period in 2018 in *N. labradorica*. This might be connected to a deeper effective living

depth in this site, exposing the specimens to a constant, low-level supply of $Ba^{2+}$ (as discussed in Sect. 4.2.2). Factors determining migration and distribution of this species comprise food supply and the bottom-water oxygenation regime (e.g., Alve and Bernhard, 1995), which conceivably could vary between the inner- and outer fjord. *Nonionellina labradorica* was less abundant in the inner fjord. Generally, its abundance has been decreasing within GF since the early 1980's, likely driven by the increase in periods of bottom water hypoxia ($[O_2] < 64$ µmol/l) (e.g., Filipsson and Nordberg, 2004). This potential

sensitivity of *N. labradorica*'s Ba/Ca signal to factors other than terrestrial input-driven Ba variability, should be considered in proxy interpretations. Oppositely, *B. marginata* is particularly promising as proxy for riverine discharge.

Incorporation of trace elements, including $Ba^{2+}$, is further known to be a function of ontogeny in some taxa, as calcification rates typically decrease and control of element partitioning increases over the growth of tests (e.g., de Nooijer et al., 2017).

Analyses across successive chambers of *B. marginata* and *N. labradorica* do show significant differences between chambers for some samples (Mann-Whitney test, Table S5). Particularly the proloculus area of *B. marginata* exhibits almost consistently higher Ba/Ca ratios than later chambers (i.e., n, n-1, n-2; Fig. S1). Because the strength and direction of the inter-chamber

correlations vary (Spearman rank correlation, Table S5) we infer a combined influence of ontogeny and seasonal differences

in pore-water [Ba] imprinting in successive chambers. As the sharp trend across the studied years is seen both in most recently

formed chambers (n), as well as prior-formed chambers (n-1, n-2; p for *B. marginata*; Fig. S1), environmental changes appear

to overprint ontogenetic effects on Ba/Ca at least across strong hydroclimatic gradients as experienced between 2018 and 2019.

**5 Implications for applications of benthic Ba/Ca as paleo-river discharge and drought proxy**

Our study highlights the potential of benthic foraminiferal Ba/Ca as indicator of drought and river discharge conditions in

near-continent records, based on signals of living foraminifera populations. We demonstrate that the input of terrestrial Ba

through riverine discharge and runoff is the dominant control on surface sediment Ba availability and foraminiferal Ba/Ca, in

the investigated coastal region. As shallow-infaunal foraminifera (e.g., *B. marginata*) show a stronger response in Ba/Ca to

continental hydroclimatic conditions, we deduce that their micro-habitat exposes them to higher $Ba^{2+}$ variability from terrestrial

sources, and therefore that these species may be more suitable as recorders of drought. The rapid cycling of bioavailable Ba at

or close to the water-sediment interface appears to be modulated by the sedimentation and dissolution of Fe-Mn oxide carrier

phases on short timescales. Therefore, we caution that a specific set of conditions may be required for the proxy to faithfully

record hydroclimatic conditions. Most importantly, GF is characterized by high rates of Fe and Mn oxide accumulation under

a generally oxic water column, which favors a strong role of dissimilatory oxide reduction in diagenesis and therefore enhanced

oxide-mediated Ba cycling. Other systems with lower oxide fluxes or persistent anoxic conditions may not be suitable to record

Ba/Ca signals of hydrodynamic changes.

In our data the variations in Ba/Ca are foremost explained by changes in physico-chemical parameters of the ambient

environment during test precipitation but seem to be partly controlled by species-specific vital effects. While the seasonal trend

in Ba/Ca across opposing river discharge conditions is significant even when pooling the two investigated taxa, we highlight

that monospecific analyses are preferable. The data set suggests that ontogenetic trends in Ba incorporation of the studied

species are probable. Nevertheless, as all investigated chambers show comparable trends in relation to continental hydroclimate

conditions, at least on annual timescales, bulk trace-elemental analyses (of specimens of restricted test sizes) are likely

sufficient for Ba/Ca proxy applications in paleo-studies.

The sensitivity of the technique to more subtle changes than experienced between the extreme and contrasting weather conditions during 2018 and 2019 remains to be determined. Further studies considering the interplay of riverine discharge, open ocean water exchange, vertical and horizontal Ba transport, and hinterland precipitation are needed to assess the ability

of the proxy to reconstruct river discharge quantitatively.

**Data availability**

The foraminiferal trace element data generated during this study will stored at the Swedish National Data Service (SND) and is made accessible to reviewers during the review process.

**Author contribution**

The concept was developed by IB and HLF. IB, TJ, KMP, MS and HFL participated in the sampling campaigns. IB, TJ, TN and KMP performed the measurements. CB, TJ and TN validated the data. IB analyzed the data and prepared the manuscript draft. All co-authors reviewed and edited the manuscript. HF administered the project, provided resources and supervision.

**Competing Interestes**

The authors declare that they have no conflict of interest.

**Acknowledgements**

The authors thank the Captain and crew of the RV *Oscar von Sydow* and RV *Skagerak* for technical assistance. We thank Sami Jokinen and Hanna Nilsson for their help during the cruises. We acknowledge the staff of the Kristineberg Marine Research Station for their support during the field campaigns. This work was supported by the Swedish Research Council VR [grant number 2017-04190]; the Crafoord Foundation; and the Royal Physiographic Society in Lund, Sweden. Tom Jilbert

acknowledges funding from the Academy of Finland (grants 317684 and 319956). The SHARK data collection is organized by the Swedish environmental monitoring program and funded by the Swedish Agency for Marine and Water Management (SwAM).

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
