# Peer review of "Drought recorded by Ba/Ca in coastal benthic foraminifera"

_Biogeosciences, 2021_

## Author Response (AR1)

Dear Dr. Cyronak,

We thank you and the two Referees for the time spent reviewing our manuscript and the constructive suggestions to improve this study. We have considered all comments and implemented changes to the manuscript accordingly. The modifications to the text and figures are detailed below as response to the comments of both referees. Please note that the line numbers refer to those in the revised version with tracked changes.

In addition to these requested changes, we have also made minor corrections, such as grammatical, and updated affiliations.

We are confident that the manuscript improved under your guidance and hope that the revised version is acceptable for publication in Biogeosciences.

Best regards,

Inda Brinkmann, on behalf of all co-authors.
* * *
Anonymous Referee 1:

This is a very good study aiming to develop benthic foraminifera as a proxy of drought using Ba/Ca ratios within the foraminiferal test. Living foraminifera have been sampled from two sites from a fjord at the Swedish west coast during three different seasons (Sept 2018, Feb 2019, June 2019) while a severe drought with reduced runoff occurred in 2018. Two species have been used for trace element analysis (*Bulimina marginata* and *Nonionellina labradorica*). The manuscript is well structured and written. Figures and tables are of high quality, and supplementary material is made available. I only have some few specific comments.

Line 1: The resulting Ba/Ca ratios reflect both river discharge and drought. Perhaps the title should be altered to "River discharge and drought recorded in..."

We are grateful to Referee 1 for the evaluation of our manuscript and appreciate the suggestions for its improvement.
We have considered the addition of 'river discharge' to the title, however, we believe 'drought' to be more precise in representing the capabilities of the foraminiferal Ba/Ca proxy as demonstrated by our study. With our data we showed a significant increase in Ba/Ca between a period of hydrological drought (i.e. year 2018) and a period of comparatively wet conditions (i.e. year 2019). On the other hand, the naming of a record of riverine discharge may be understood as a quantitative measure of hydrological conditions. As discussed in the manuscript, we suggest the proxy to have potential in qualitatively tracing paleo-drought periods, whereas more subtle changes in riverine discharge may not be resolvable by foraminiferal Ba/Ca records unless further studies resolve uncertainties regarding hinterland precipitation, terrestrial and riverine Ba concentrations and horizontal Ba transport processes.

Line 52 [now line 55]: The review by Howe et al. (2010) is not specifically on benthic foraminifera. It would be beneficial to have one to two additional references here specifically on foraminifera, e.g. Murray (2006), ISBN 0521828392.

We have added Murray (2006) as reference.

Lines 244-246 [now lines 250–252]: Ba/Ca ratios represent river discharge 30 or 60 - 90 days ago. Are there other factors causing this than foraminiferal growth and lag in the transport of Ba/Ca? What about hydrological fluctuations of the river or other environmental factors?

Thank you for this suggestions. We have reformulated our explanation of the potential lag time between riverine Ba input and benthic foraminiferal incorporation, and further added a reference to section 4.2 (lines 252–255), where several environmental factors of potential influence on water-column Ba transport are discussed in depth. Further, we have added a statement suggesting the potential of varying riverine Ba concentrations, which may

affect Ba supply to the fjord, although we regard this to be negligable in face of the contrasting hydrological conditions having persisted in 2018 and 2019, respectively (lines 261–264)

Lines 260-261 [now lines 273–274]: "... with generally lower Ba/Ca in the outer-fjord site during 2018 (Fig. 3)". The values are relatively similar between sites for *N. labradorica*? Please make describe this in the text instead of using "generally".

We reformulated the sentence in order to clarify and highlight the difference between the two species (lines 273–275).

Line 382-405 [now 398 and following]: In the section on implications, it would be beneficial to evaluate the possible effects of the lag of 30 or 60-90 days between the river discharge and the Ba/Ca ratios when establishing paleorecords of river discharge and possibly drought.

We have added a sentence cautioning against interpretations of foraminiferal Ba/Ca with regards to interpretations of hydro-dynamic changes on short timescales (e.g., sub-annual) in lines 417–419: "High-temporal resolution interpretation of hydrodynamic changes in paleo-applications (e.g., sub-annual timescales) should be considered with caution unless uncertainties regarding the duration of Ba down-ward transport in the water-column and lag between Ba input and foraminiferal incorporation can be resolved."

Anonymous Referee 2:

This paper addresses the potential of using Ba/Ca on benthic foraminifera as a proxy for hydrological changes based on a case study from the Gullmar Fjord, Sweden. The results of the study indicate a potential for using especially Ba/Ca of B. marginata as an hydrological proxy, despite some caveats regarding  specific conditions prevailing at the study sites and other factors that might influence the foraminiferal ratio. The paper is very well written und the figures are of high quality. The motivation of the study, experimental design and the results are very concisely presented and the data is thoroughly discussed including its limitations. In summary, I really enjoyed reading the manuscript and support its publication in Biogeosciences.

However, there is one important aspect that appears to be missing: According to the Material & Methods section, water samples have been obtained during the sampling campaigns, but no trace element analyses on the water samples are reported. This might be an important information, as the pore water profiles show no distinction between the different sampling periods, despite the presumed strong hydrological difference between the dry year 2018 and the wet year 2019. Hence, the different Ba/Ca signature of B. marginata between both years does likely not derive from differences in the pore water composition. Albeit bottom water Ba/Ca would also provide only snapshots, it might provide important information about the Ba-transfer into the pore water and the general amplitude of bottom-water Ba/Ca fluctuations in the Fjord. The authors use the hydrological observational data to indirectly infer Ba/Ca in the Fjord's water column but direct measurement of water composition might well be helpful. Hence, I would like the authors to comment if such measurements exists or add them if possible as water analyses are analytically relatively straight forward.

We thank Referee 2 for the assessment of our manuscript and constructive feedback. We agree that measurements of Ba throughout the water column may directly link the input and transport of riverine Ba to the foraminiferal response in Ba/Ca. Data points of both dissolved and solid-phase Ba throughout the water-column may have the potential to shed light on our hypothesis of oxide-mediated Ba shuttling in the fjord. Further, tracing Ba throughout the water-column on a temporal scale may inform about Ba transport duration and thus could resolve uncertainties about the 'lag' between hydrological signal and foraminiferal Ba/Ca response. Ideally, such data would have been collected both during a drought period, and before, during and after a rainfall event. Unfortunately, we did not collect and store water samples beyond those that were geochemically analysed as part of the project and thus cannot close this observational gap. We did, however, measure dissolved Ba concentrations in water from c. 1 cm above the water-sediment interface (e.g. Fig. 4), although we cannot exclude a significant impact of sediment geochemical cycling on those values due to the close proximity to the sediment surface. These bottom-water values showed no significant differences between our three sampling points, likely due to 'snapshot' character of our measurements, as discussed in section 4.2.

Further specific comments:

- The supplementary information should be separated for figures and tables. Supplementary Figure 4 has no caption.

We have separated the supplementary information. All figures are described by a caption.

- Fig. 2a: please indicate the location of the study area.

Yes, we added a rectangle to the maps to indicate our study area.